# *Corona was scary, lockdown was worse*: A mixed-methods study of community perceptions on COVID-19 from urban informal settlements of Mumbai

**Sudha Ramani** , **Manjula Bahuguna, Apurva Tiwari, Sushma Shende, Anagha Waingankar, Rama Sridhar, Nikhat Shaikh, Sushmita Das, Shanti Pantvaidya, Armida Fernandez, Anuja Jayaraman** *

SNEHA (Society for Nutrition, Education and Health Action), Mumbai, Maharashtra, India

* anuja.jayaraman@gmail.com

**Data Availability Statement:** As per the data sharing policies of our organization, SNEHA, program datasets can be shared online only after

## Abstract

The COVID-19 pandemic has magnified the multiple vulnerabilities of people living in urban informal settlements globally. To bring community voices from such settlements to the center of COVID-19 response strategies, we undertook a study in the urban informal settlements of Dharavi, Mumbai, from September 2020-April 2021. In this study, we have examined the awareness, attitudes, reported practices, and some broader experiences of the community in Dharavi with respect to COVID-19. We have used a mixed-methods approach, that included a cross-sectional survey of 468 people, and in-depth interviews and focus group discussions with 49 people living in this area. Data was collected via a mix of phone and face-to-face interviews. We have presented here the descriptive statistics from the survey and the key themes that emerged from our qualitative data. People reported high levels of knowledge about COVID-19, with television (90%), family and friends (56%), and social media (47%) being the main sources of information. The knowledge people had, however, was not free of misconceptions and fear; people were scared of being forcefully quarantined and dying alone during the early days of COVID-19. These fears had negative repercussions in the form of patient-related stigma and hesitancy in seeking healthcare. A year into the pandemic, however, people reported a shift in attitudes from 'extreme fear to low fear' (67% reported perceiving low/no COVID risk in October 2020), contributing to a general laxity in following COVID-appropriate behaviors. Currently, the community is immensely concerned about the revival of livelihoods, that have been adversely impacted due to the lockdown in 2020 as well as the continued 'othering' of Dharavi for being a COVID hotspot. These findings suggest that urban informal settlements like Dharavi need community-level messaging that counters misinformation and denial of the outbreak; local reinforcement of COVID-appropriate behaviours; and long-term social protection measures.

three years of completion of the program or project. Data cannot be shared publicly because it contains information that can potentially identify participants and compromise anonymity. The authors can share data tables (survey) and anonymised notes (qualitative interviews) on request. This qualifies as the minimal data set underlying our study. The Chief Executive Officer, Ms. Vanessa D'Souza (vanessa@snehamumbai. org) will be a non-author point of contact for data access.

**Funding:** This research was funded through the Epic Foundation and Give Foundation, as a part of larger implementation grants. The funders had no role in study design, data collection and analysis, decision to publish, or preparation of the manuscript.

**Competing interests:** The authors have declared that no competing interests exist.

## Introduction

Urban informal settlements are home to more than one billion people across the globe and comprise about one-third of the world's urban population [1]. These areas are extremely vulnerable to 'shocks' of all kinds, be it natural disasters, epidemics, or financial crises [2–4]. The 'shock' due to COVID-19 has been no exception. Issues of space constraints, crowding, the lack of sanitation and water, and the use of shared communal spaces like public toilets make the implementation of COVID-19 mitigation interventions very challenging in these areas [5, 6]. For instance, the lack of space renders interventions such as 'social distancing' and 'self-quarantining' inconsequential in practice [5]. In addition, the lockdown measures initiated by many countries in response to COVID-19 outbreaks appear to have contributed to income loss, food insecurity, and reduced access to health services, further exacerbating the vulnerability of people living in such settlements [2, 3]. All this has led to a renewed recognition that a successful response to COVID-19 in these spaces needs a deep understanding of the unique, context-specific needs of communities that live there [2, 7, 8].

In the past year, the COVID-19 pandemic has spurred a range of insightful Knowledge, Attitude and Practices (KAP) surveys in communities across countries in both urban and rural geographies [9–13]. Yet, very limited literature reports on the experiences of communities from urban informal settlements of the global south. We found only a handful of studies—two KAP surveys from Kenya [14, 15], one study that compiled stakeholder opinions from four LMICs [6], and one KAP survey from Bangladesh [16] that report specifically on community experiences from urban informal settlements during COVID-19. Given the uniqueness of these geographical spaces and their vulnerability to COVID-19, there is clearly a need for more in-depth scholarship that brings to the forefront voices of the community, so as to facilitate reflections on the pandemic response in these areas.

This KAP study is based in Dharavi, Mumbai city, in the state of Maharashtra (see Fig 1 for location). As is the case in India, Mumbai is also reaping the benefit of demographic dividend where over 70% of Mumbai's total population is of working age (15 to 64 years) population [17]. Dharavi is one of the biggest urban informal settlements in Asia covering an area of 2.1 km². It is considered one of the most densely populated places in the world with a population of nearly 1 million [18]. It is home to around 5000 small-scale enterprises and 15000 single-room factories of leather, pottery, and textiles [18]. The livelihood opportunities in Mumbai attract migrants to Dharavi from different parts of India [19] and a large majority of people living in these informal settlements belong to the working-age group [20, 21]. On average 8–10 people live in small one-room houses that are situated very close to each other in narrow lanes. A recent paper notes that around 70% of the population in Dharavi use common toilets and water facilities [22].

Dharavi reported its first case of COVID-19 on April 1st, 2020 and by the end of the month emerged as a hotspot with 491 cases and a 12% monthly growth rate in cases [23]. In the first two weeks of May 2020, a sharp increase in cases was observed. This was followed by a downward trend in the number of cases until Jan 2021 [23, 24]. Between April 1st, 2020, and June 15th, 2020, a total of 2068 cases were reported of which 1040 people recovered and 77 people died [24]. Fig 2 depicts the numbers of COVID-19 cases in Dharavi between April 2020-April 2021.

The initial rapid increase of cases in Dharavi in the months of April-May 2020 elicited an intensive response from the government in this area [25]. The response to COVID-19 in Dharavi (which came to be known as the 'Dharavi model' for combating COVID) has been touted

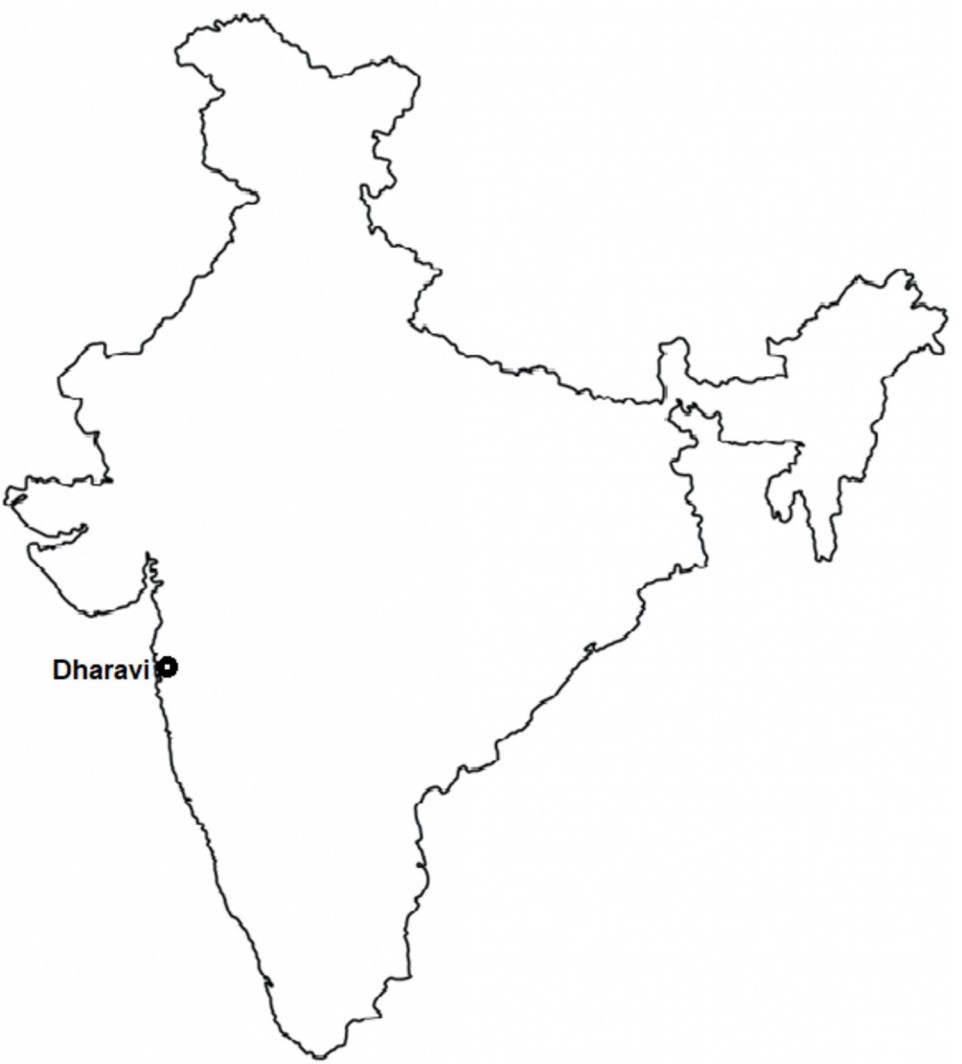

**Fig 1. Map of India showing the location of Dharavi.**

widely as a template for 'flattening the curve' against COVID-19 in informal settlements [25–27]. Some details of the Dharavi response have been given in Box 1.

Author compilation using sources:

1. Public Health Department, Government of Maharashtra Novel Corona Virus-Government of Maharashtra Public Health Department, India

2. Press Release, Ministry of Home Affairs, Government of India MHA Press Releases | Ministry of Home Affairs | Government of India

3. Municipal Corporation of Greater Mumbai data on Dharavi COVID-19 cases

While attention has been given to the epidemiological situation of COVID-19 and the government response strategy in Dharavi, the experiences of the community living there have not been documented so far. In fact, we could not find any published community studies from

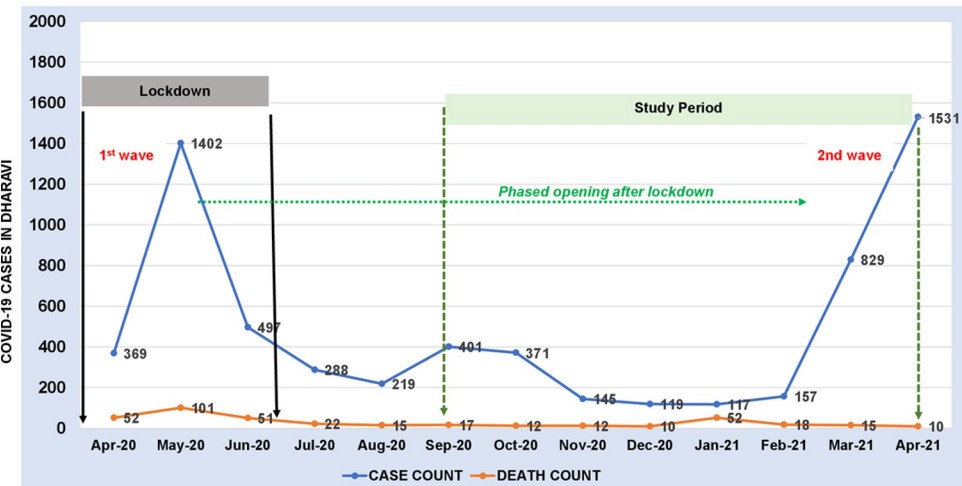

**Fig 2. COVID-19 timeline in Dharavi and our study period.**

urban informal settlements in India on COVID-19; despite the fact that over 65 million (17.4%) people live in urban informal settlements in the country [29, 30].

In this study, we have attempted to provide insights into the perceptions of the community in Dharavi. In specific, this study aims to

1. Examine the Knowledge, Attitudes and Practices (KAP) of the Dharavi community with respect to COVID-19 and

2. Document some of the broader experiences of the community during the first wave of the COVID-19 outbreak.

The study was done following the first wave of the COVID-19 outbreak in India and before the second wave, with data collected between September 2020-January 2021 (see Fig 2). During the time of our data collection, vaccination of the general population had not begun in India.

## Box 1. The response to COVID-19 in Dharavi

Dharavi has been praised widely for its pro-active response to the first wave of COVID-19 in mid-2020. From March 2020, intensive efforts were made by the local municipal administration to control the spread of COVID-19- including efforts directed at screening and testing, conduction of fever camps, regular disinfection of shared toilets and other public spaces, and strict prohibition of gatherings including curfews [25, 26]. If found positive, people were quarantined in local centers, set up specifically for this purpose.

About 700000 people were screened for symptoms, suspected 14000 were tested, and 13000 were placed in quarantine centers [27]. The local administration also partnered with private doctors and non-governmental organizations to bolster the response. These efforts had complemented the nationwide lockdown- that restricted movement and economic activities pan India—between 21st March to 31st May 2020 [28]. All these measures together appeared to bring the rising cases in Dharavi under control as the growth rate reduced to 1.02% in June 2020 from 4.3% in May 2020 [23] (see Fig 2).

This study combines survey methods with in-depth qualitative inquiry in the community. Due to the strict lockdowns and travel restrictions, there have been very few qualitative studies that have been conducted in this time period that provide rich evidence-based insights into the COVID-19 outbreak [31, 32]. So far, we could find one in-depth study from urban informal settlements that explores community experiences during COVID-19 through qualitative methods [6]. The combination of qualitative and quantitative methods in this study permits a deeper, more integrated understanding of the topic being examined here [33].

## Methodology

This is a mixed-methods study that combines both quantitative and qualitative approaches. For this study, we have collected data from the field site of the Non-Government Organization (NGO), the Society for Nutrition Education and Health Action (SNEHA) from three administrative divisions (for the survey) and two administrative divisions (for in-depth discussions) from Dharavi. SNEHA has been working in Dharavi on issues pertaining to maternal, adolescent, and child health and the prevention of violence against women and children for over 10 years. SNEHA, in close coordination with public health systems, works at the community level to improve the accessibility of health services to the most vulnerable sections of society. This study has been done as a part of our larger programmatic efforts intended to combat COVID-19 in this area, through a range of activities including building community partnerships, intensive messaging, and direct relief work. As an NGO, SNEHA has a long-standing relationship with the community in Dharavi, and its field staff and community volunteers helped us in identifying and recruiting participants for the study. More methodological details are given below.

### Quantitative methods

We conducted a cross-sectional needs assessment survey between September to October 2020. The main objective of the survey was to assess the level of COVID-19 awareness and preventive practices of people in the community. Due to movement restrictions in Mumbai, we conducted telephonic interviews instead of face-to-face interviews. The list of phone numbers was gathered from two sources. 1). Frontline health workers of a public nutrition scheme (Integrated Child Development Services) and 2). Community volunteers who were associated with SNEHA. A total of 6100 phone numbers of people who were 18 years and older was collected from nearly 15000 families residing in three administrative divisions of Dharavi. From this list, a random sample of phone numbers was selected. Following Austrian et al. (2020), we estimated a sample size of a minimum of 382 participants with a margin of error of +/-5% at a 95% confidence level from a conservative prevalence estimate of 50% [15]. Based on our experience of other surveys in the area, a 15% non-response rate for refusals and erroneous data was also added to arrive at the final sample size of 450.

**Survey questionnaire.** A 38-item questionnaire was adapted from existing survey tools on capturing behavioural insights on COVID-19 from the community [15, 34]. In addition to the basic demographic information, the survey included questions on COVID-19 related to knowledge, perceived risk, perceived efficacy of preventive measures, current practices to prevent infection, sources of information, and awareness of government helpline numbers and activities. The section capturing knowledge and awareness of COVID-19 covered three aspects of the disease; mode of spread, symptoms, and prevention practices. In response to the question "How does the coronavirus spread?", the tool had options to capture misinformation as well (for instance, there was an option that the virus spreads through food). Similarly, in response to the question "How to prevent coronavirus?", the tool had options to capture

misinformation (for example, we asked about the use of herbal/home remedies, avoiding contact with animals, and sleeping under mosquito nets). We have appended the survey tool to this paper for further reference (see supplementary information).

We also considered two subjective dimensions- the perceived risk of COVID-19 and the perceived efficacy of recommended preventive actions- as determinants of attitudes towards COVID-19. We measured the perceived risk of COVID-19 infection by asking a question on the likelihood of getting infected with COVID-19 and in cases with low-risk perception, we asked about the reasons for the same. Perceived efficacy of recommended preventive actions was captured by a question on the effectiveness of the preventive actions at reducing the risk of COVID-19. Detailed information on mask-wearing (type of mask, when to wear, sharing of masks, washing frequency of reusable masks), handwashing practices (when to wash hands), and physical distancing were captured. Protective behaviour exhibited by the participants in the past one week for mask-wearing and the last 24 hours for handwashing was also captured. In addition, we collected information on awareness and use of different helpline numbers (COVID-19, food distribution, mental health, domestic violence, sexual abuse), COVID-19 related activities by the local administration, and food supplies. Finally, data on information needs and worries of the participants were also collected to strengthen SNEHA program strategies.

**Data collection.**    A well-trained team comprising of 6 investigators and 1 supervisor conducted the survey. Multiple online mock interviews were first conducted to understand any difficulty in tool administration before the final survey was administered. For questions with multiple choices in answers, investigators did not read out all options to the participants rather cautiously probed for their views. Each investigator was given a list of randomly selected contact numbers for interviews. If investigators were unable to establish contact with the potential participant after making 3 calls per day for 3 consecutive days, then the household was marked as closed/respondent not available, and the next household number from the list was selected for interview. Reasons for non-participation included numbers being switched off or not reachable, or the people having migrated elsewhere. Prior to the interview, informed oral consent was obtained from all participants. Each interview lasted for about 20 to 25 minutes. Attempts were made to take appointments to call the participants at their convenience for the survey. Data was collected using CommCare, an open-source mobile-based application. The application had automatic skips and validation constraints which minimized errors and maintained data quality.

**Analysis.**    Identifiable fields were removed from the final dataset and STATA version 14.0 was used for analysis. We used descriptive statistics to summarize the data and variables are presented using frequency distribution, percentage, and mean.

## Qualitative methods

We recruited participants for the study by beginning with the phone number lists that we already had from the survey. We reached out to others by snowballing from these lists to strengthen the evidence on particular themes iteratively. We interacted with a total of 49 participants from the community in Dharavi.

We initially conducted 16 phone interviews with community members in Dharavi over the phone, diversified in terms of gender and age; 3 interviews with patients who had recovered from COVID-19; and 6 interviews with community volunteers who had helped with relief work during the pandemic. After the first round of analysis, we felt that a few themes (such as those related to recent migrants who were a particularly vulnerable sub-group) were missing- since many of them were not accessible by phone. Thus, we went to Dharavi and conducted 9

(8 short, 1 in-depth) additional face-to-face interviews with people who did not have access to phones or had gone back to their villages during the lockdown and had returned seeking jobs. In addition to this, we conducted 3 face-to-face Focus Group Discussions (FGDs) in the community. These FGDs were used to validate some of our preliminary lines of thinking and explore certain themes from the interviews in depth.

We used a guide to help focus our questions; the key themes from which are given in Table 1. The interviews and FGDs were conducted in a mix of local languages, Hindi and Marathi. Interviews were conducted by the authors MB, SR, RS, and NS and ranged between 25 minutes to an hour. All interviewers were well-trained in qualitative data collection and analysis methods and familiar with the Dharavi context. The interviews were recorded, translated, and transcribed into English. We did not transcribe the FGDs but used information from these written in the form of detailed field notes.

On average, only 2 of 5 people we tried to contact over the phone agreed to participate. We sometimes had to call multiple times to schedule and fix interviews at the convenience of community participants. Reasons for non-participation included reported lack of time, inconvenience, participants not being available at the time planned for the interview, and people not picking up their phones. Also, we felt that people were hesitant to talk about COVID-19 due to a strong sense of stigma in Dharavi that surrounded the disease. We did not offer any compensation to the participants for talking to us.

As customary to qualitative studies, data collection and analysis were done iteratively. We used generic thematic analysis techniques to help analyze our data; in this type of analysis, data is first sorted and organized into themes, further summarized as data displays (tables, case studies, and conceptual figures), and then interpreted [35]. The quantitative survey was done prior to the qualitative study; hence we could use our interviews to verify and provide depth to some of the trends that emerged from the survey (for instance, our survey results suggested high-levels of mask-wearing and our interviews supplemented this information by trying to understand in-depth why such high levels were reported).

To analyze the interviews, we used 'framework analysis' wherein we constructed a matrix in MS excel that helped to compare cases across themes [35, 36]. For doing this, we started with a broad codebook in which initial themes for the analysis (derived from the survey and guide) were noted. We constructed a matrix with cases (interviews with participants) as rows, and the various themes in the codebook as columns. These columns were not fixed but modified as the data emerged. Two of the researchers read the transcripts and coded these separately. Preliminary ideas that emerged from the transcripts were discussed in a group. We held weekly debriefs as a team in the months of December 2020-January 2021 to discuss emerging ideas and lines of thinking in a group, and iteratively build on the themes that emerged. We used the qualitative software NVivo version 10 as well as MS word to aid the coding process.

We had conducted three face-to-face FGDs to validate the key findings from the interviews; such member-checks have been considered crucial to triangulate perspectives and establish the credibility of ideas [37]. During these FGDs, we shared the key findings of the interviews with participants and checked for the resonance of these findings with peoples' experiences. We also used FGDs to deepen some themes that we felt were missing in the interviews. For instance, our interviews lacked strong evidence on gender differences in the perception of the pandemic; and we used the FGDs to probe more on these lines. We finally synthesized and presented the results of the study thematically.

Ethical approval for the study was obtained from the Institutional Ethics Committee, the Bandra Holy Family Hospital & Medical Research Centre, Mumbai. Recorded oral consent was taken from all participants.

**Table 1. Details of the qualitative data collection.**

| Participants | Community members | | | COVID-19 recovered individuals | Community volunteers |
|---|---|---|---|---|---|
| Methods | In-depth interviews (17)* | Short field interviews# (8) | Focus group discussions^ (3) | In-depth interviews (3) | In-depth interviews (6) |
| **Total number of participants (49)** | 17 | 8 | 15 | 3 | 6 |
| | | | *FGD1 (female) - 7* | | |
| | | | *FGD2 (male) - 5* | | |
| | | | *FGD3 (migrant workers) - 3* | | |
| **Age in years** | | | | | |
| 18–30 | 6 | 3 | 2 | 0 | 2 |
| 31–40 | 3 | 2 | 7 | 1 | 2 |
| 41–50 | 4 | 1 | 6 | 1 | 2 |
| 51 and above | 4 | 2 | 0 | 1 | 0 |
| **Sex** | | | | | |
| Male | 9 | 0 | 8 | 2 | 1 |
| Female | 8 | 8 | 7 | 1 | 5 |
| **Religion** | | | | | |
| Hindu | 14 | 4 | | 3 | 4 |
| Muslim | 3 | 4 | | 0 | 2 |
| **Occupation** | | | | | |
| Housewife/unemployed | 9 | 7 | 7 | 0 | 3 |
| Unskilled job | 3 | 1 | 5 | 1 | 1 |
| Skilled job | 3 | 0 | 3 | 1 | 2 |
| Retired | 2 | 0 | 0 | 1 | |
| **Resident of Dharavi** | | | | | |
| By birth/more than 25 years | 8 | - | - | 3 | 1 |
| 11–25 years | 5 | - | - | 0 | 1 |
| 5–10 years | 3 | - | - | 0 | 3 |
| Less than 5 years | 1 | - | - | 0 | 1 |
| **Volunteer with SNEHA** | | | | | |
| Less than one year | - | - | - | - | 2 |
| More than one year | - | - | - | - | 4 |

**Key themes in the tools used for community discussions**
- Knowledge and awareness related to COVID-19
- Shift in attitude towards COVID-19
- Prevention measures in Dharavi
- Trust in the government response
- Access to health and other public services during COVID-19
- Othering of the Dharavi community- stigma and discrimination
- Challenges faced due to COVID-19

*16 telephonic and 1 face to face interview with a male migrant during field visit.

#Short interviews were conducted to supplement the online interviews and reach people who were not accessible over the phone. Interviews were kept short to minimize researcher contact. It included conversations with a diverse group of people such as a local shop keeper, a migrant family, and a mother who had lost her 8 years old child to cancer during lockdown.

^Intended to validate some of the findings from the in-depth community interviews.

# Results

We report the findings of this study under two sections. Section 1 pertains to the KAP questions related to COVID-19 that we asked in the survey. Section 2 reports on our qualitative findings.

### Section 1: Quantitative findings

**Socio-demographic characteristics of the survey participants.** A total of 468 respondents participated in the study. The mean age of the participants was 33.2 ± 9.6 years and ranged from 18 to 77 years. Nearly half (47.4%) of the respondents were in the age group of 18–30 years in line with the demographic characteristics of the population in Dharavi. The average household size was 5.3 ± 2.3 people. Only one-fourth of the respondents reported having access to a private toilet facility in their households while the majority (74.8%) reported using public/shared toilets (Table 2).

**Sources of information on COVID-19.** Our findings showed that 90.2% of people reported television and radio as the main source of information, followed by family or friends (56.4%) and social media (47.4%). Health system workers and NGO workers were mentioned as sources of information on COVID-19 by fewer respondents. However, we also asked a question on which sources of information were most trusted by people. In response to this question, we found that the trust reported by respondents on information received from health systems and NGO workers was high (96–97%). At the same time, close to 60% of the people reported trusting the information that they had obtained from social media (Table 3).

**Knowledge related to COVID-19 spread, symptoms, and prevention.** More than three-fourths (76.3%) of the participants knew that infected people could transmit the COVID-19 virus and 68.8% were aware that the transmission could occur through droplets. Few (16.7%) also mentioned that the disease was airborne. The spread of COVID-19 by eating contaminated food, and drinking unclean water was also reported by 5% of the respondents. A large majority of the participants were aware of the symptoms like dry cough (89.1%) and fever or chills (87.6%). Regarding awareness about preventive measures, most people reported washing hands or using sanitizer regularly (97.4%), wearing face masks (93.4%) and maintaining social distance or staying indoors (84.8%) as the main protective behaviours against COVID-19. Some people also reported the use of herbal medicines or home remedies (40.4%), and drinking treated or boiled water (36.3%) as preventive measures (Table 4).

**Attitude and practices pertaining to COVID-19.** A quarter of the respondents in our study reported that they perceived no risk from COVID-19, while 42% felt that they were at

**Table 2. Socio-demographic characteristics of the survey participants among Dharavi residents, Mumbai, India (N = 468).**

| Age in years | n | % |
|---|---|---|
| 18–30 | 222 | 47.4 |
| 31–40 | 168 | 36.0 |
| 41–50 | 49 | 10.5 |
| 51 and above | 29 | 6.2 |
| Mean age (±SD) | 33.2 (± 9.6) | |
| **Sex** | | |
| Male | 226 | 48.3 |
| Female | 242 | 51.7 |
| **Household size** | | |
| 1–5 | 292 | 62.4 |
| 6 and above | 176 | 37.6 |
| Mean household size (±SD) | 5.3 (±2.3) | |
| **Toilet facility** | | |
| Public/shared | 350 | 74.8 |
| Private | 118 | 25.2 |

**Table 3. Sources of information and their trustworthiness in survey participants among Dharavi residents, Mumbai, India.**

| Sources of information* | Source of information (N = 468) | | Trust on source | |
|---|---|---|---|---|
| | n | % | n | % |
| Electronic media | 422 | 90.2 | 339 | 80.3 |
| Family/friends | 264 | 56.4 | 223 | 84.4 |
| Social media | 222 | 47.4 | 129 | 58.1 |
| Health systems (doctors and frontline workers) | 198 | 42.3 | 191 | 96.4 |
| Non-Government organizations | 104 | 22.2 | 101 | 97.1 |
| Public events | 80 | 17.0 | 69 | 86.2 |
| Community leaders | 12 | 2.5 | 10 | 83.3 |

*Questions allow respondents to choose multiple responses.

low risk. The perception of being at no risk was primarily based on the practices of wearing masks (76%), washing hands (76%), and following social distancing (61.5%). A few participants also reported reasons like "I am young and healthy" (13.7%) and "God protects me" (11.1%) as reasons for perceiving themselves at no risk. The majority (97.2%) believed in the efficacy of adopting COVID-19 appropriate behaviours. When asked about the preventive practices observed by participants in the week preceding the survey, people mentioned

**Table 4. COVID-19 knowledge (spread, symptoms and prevention) of survey participants among Dharavi residents, Mumbai, India (N = 468).**

| | | Participant's responses | n | % |
|---|---|---|---|---|
| Knowledge | Spread* | Direct contact with an infected person | 357 | 76.3 |
| | | Droplets from aninfected person | 322 | 68.8 |
| | | Touching contaminated surfaces/objects | 186 | 39.7 |
| | | Airborne | 78 | 16.7 |
| | | Eating contaminated foods | 25 | 5.3 |
| | | Drinking unclean water | 24 | 5.1 |
| | Symptoms* | Dry cough | 417 | 89.1 |
| | | Fever/chills | 410 | 87.6 |
| | | Congestion/runny nose | 175 | 37.4 |
| | | Shortness of breath/difficulty in breathing | 246 | 52.6 |
| | | Sore throat | 211 | 45.1 |
| | | Muscle/body ache | 133 | 28.4 |
| | | Headache | 132 | 28.2 |
| | | Tiredness/fatigue | 86 | 18.4 |
| | | Loss of taste or smell | 15 | 3.2 |
| | | Chest pain or pressure | 9 | 1.9 |
| | Prevention* | Wash hands regularly using sanitizer or soap/water | 456 | 97.4 |
| | | Wear face masks | 437 | 93.4 |
| | | Social distancing/staying indoors | 397 | 84.8 |
| | | Use herbal/ayurvedic/home remedies | 189 | 40.4 |
| | | Drink only treated or boiled water | 170 | 36.3 |
| | | Cover mouth and nose when coughing or sneezing | 85 | 18.2 |
| | | Avoid close contact with anyone who has a fever and cough | 57 | 12.2 |

*Questions allow respondents to choose multiple responses.

**Table 5. COVID-19 related attitude and practices of survey participants among Dharavi residents, Mumbai, India (N = 468).**

| | Participant's responses | n | % |
|---|---|---|---|
| **Attitude** | **Perceived risk of COVID-19** | | |
| | Low risk | 197 | 42.0 |
| | No risk | 117 | 25.0 |
| | High risk | 94 | 20.0 |
| | Don't know | 60 | 12.8 |
| | ***Reasons for "no risk at all" (N = 117)**** | | |
| | I wear a face mask | 89 | 76.0 |
| | I wash hands/use sanitizer | 89 | 76.0 |
| | I practice social distancing | 72 | 61.5 |
| | I stay at home | 28 | 23.9 |
| | I am young and healthy | 16 | 13.7 |
| | God protects me | 13 | 11.1 |
| | I adhere to government guidelines | 9 | 7.7 |
| | **Perceived efficacy of protective behaviours** | | |
| | Somewhat effective | 249 | 53.2 |
| | Very effective | 206 | 44.0 |
| | Not effective at all | 13 | 2.8 |
| **Practices#** | **Preventive practices in last 1 week*** | | |
| | Wash hands regularly using soap and water/alcohol sanitizer | 460 | 98.3 |
| | Wear masks if going outdoors | 445 | 95.0 |
| | Social distancing/staying indoors | 388 | 82.9 |
| | Herbal medicines/home remedies | 342 | 73.0 |
| | Disinfecting surfaces | 99 | 21.2 |
| | ***Type of masks**** | | |
| | Reusable cloth mask | 376 | 80.3 |
| | Disposable medical mask | 75 | 16.0 |
| | Traditional scarf/cloth piece | 40 | 8.5 |

*Questions allow respondents to choose multiple responses.

# We did not have an option on ventilation as a practice.

washing hands (98.3%), wearing masks (95%), maintaining social distance (82.9%), and disinfecting surfaces (21.2%). Nearly three-fourths of the participants reported the use of herbal medicines or home remedies as preventive measures. The majority of the participants (80.3%) reported using reusable cloth masks and only 16% reported the use of medical masks (surgical or N-95 masks) (Table 5).

**COVID-19 response activities by the local administration in Dharavi.** The disinfection of the public toilets (66.9%), distribution of essential items (64.1%), and guarding of the containment area (53.6%) were mentioned by participants as the main activities carried out by the local administration. Over half of the respondents reported that door-to-door screening for COVID-19 cases was done by the government. The majority (84.8%) of the participants purchased groceries from government-subsidized stores. Data indicated that only 35.2% of the participants were aware of the available government helplines for COVID-19 testing and food distribution. Further, only 4.2% reported having ever used these helplines (Table 6).

**Worries and information needs of the survey participants.** Nearly half, 44% of the respondents mentioned unemployment as their major worry at the time of the survey and a

**Table 6. COVID-19 response activities by the local administration as reported by survey participants among Dharavi residents, Mumbai, India (N = 468).**

| COVID-19 response activities* | n | % |
|---|---|---|
| Disinfection of public toilets | 313 | 66.9 |
| Distribution of essentials items (groceries/medicines) | 300 | 64.1 |
| Guarding of the containment area | 251 | 53.6 |
| Conducting a door-to-door COVID-19 screening | 242 | 51.7 |
| Disinfection of the area after positive case finding | 221 | 47.2 |
| Communication about prevention and quarantine | 142 | 30.3 |
| Testing suspected cases of COVID-19 and their contacts | 76 | 16.2 |
| Quarantine infected patients and their contacts | 63 | 13.4 |
| Testing senior citizens or high-risk people | 56 | 12.0 |
| **Food supplies during lockdown*** | | |
| Purchased from government-subsidized stores | 397 | 84.8 |
| Got it for free from government-subsidized stores | 275 | 58.8 |
| Received cooked food/ration from ICDS[a] or employers | 209 | 44.6 |
| Non-governmental organizations | 160 | 34.2 |
| **Government helplines*** | | |
| Awareness about helpline number (COVID-19 testing, food distribution) | 165 | 35.2 |
| Ever used helpline numbers | 20 | 4.2 |

*Questions allow respondents to choose multiple responses.

[a] The Integrated Child Development Services (ICDS) is a government scheme that provides supplementary nutrition in form of take-home rations to children 6 months to 6 years, pregnant women, and lactating mothers.

few (26.3%) also reported an inability to pay bills. Personal health (14.9%) and family health (28%) were the main concern of some, and a few participants shared worries related to food supplies (13%) and liberty of movement (9.8%) as well. More than half (54%) of the participants said that they did not need any further information on COVID-19 (Table 7).

## Section 2: Qualitative findings

In this section, we report on attitudes of people towards COVID-19; preventive practices, and the beliefs of respondents on the efficacy of these practices. We also elucidate community experiences during the lockdown period and after, and the repercussions that the lockdown had in Dharavi.

**Despite awareness about COVID-19, myths and rumours prevailed.** COVID-19 was often referred to as "carona" in the community and all respondents we spoke to had heard of the disease. In line with the survey results shown in Table 4, discussions with the community showed that they were well-aware of COVID-19 transmission mechanisms, symptoms, and preventive measures.

When we asked the community about which sections of the population (male, female, elderly) were more vulnerable to COVID-19, people generally responded that everybody was at risk. People did not mention any differences between men and women in terms of the risk of COVID-19 infection. Many respondents, however, mentioned that elderly people, pregnant women, and people with co-morbidities were at higher risk. People in the community also reported being protective of their children and described them as one of the "high-risk" categories. Low immunity was stated as the main reason for the vulnerability of children as well as the elderly.

**Table 7. Worries and information needs of the survey participants among Dharavi residents, Mumbai, India (N = 468).**

|  | n | % |
|---|---|---|
| **Major worries**[*] |  |  |
| Unemployment | 206 | 44.0 |
| No worries | 160 | 34.2 |
| Family's health | 131 | 28.0 |
| Inability to pay bills | 123 | 26.3 |
| Personal health | 70 | 14.9 |
| Restricted access to food supplies | 61 | 13.0 |
| Restricted liberty of movement | 46 | 9.8 |
| **Information needs**[*] |  |  |
| No need | 253 | 54.0 |
| Protection from COVID-19 | 134 | 26.5 |
| Economic impact of COVID-19 | 36 | 7.7 |
| Children's education | 14 | 3.0 |
| Food/grocery supplies | 14 | 3.0 |

[*]Questions allow respondents to choose multiple responses.

*"Children younger than 10 years are weak and we should take care of them. The old people living in my house can also get affected since they are very fragile"* (**Female, 31 years, housewife with one child, Dharavi resident for 13 years**)

*"Yes, people say, carona affects according to age. Old people's immunity power is less and young children also because of their age, their immunity is low."* (**Male, 25 years, works in a hotel, living in Dharavi since birth**)

*"Those, who have high sugar, high BP, those who can't breathe, they face more problems. They are mostly affected. . . Older people are more vulnerable to carona. Older people, children etc. Pregnant women are also vulnerable."* (**Female, 44 years, nurse aid, currently**

unemployed, Dharavi resident for 24 years)

Similar to our quantitative data on sources of information (Table 3), we found that television news, family and neighbours, as well as social media, were reported as important sources of information on COVID-19. In addition, schools and workplaces were also mentioned as sources (these two options were missing in information from our survey). The availability of information through different sources appeared to have enabled high levels of awareness of the pandemic. But information from some of these sources (particularly social media, information from neighbours) also seemed to play a role in perpetuating certain rumours and myths. For instance, some people believed that only the 'rich' get affected by COVID-19. In addition to the usual ways of transmission, eating meat was considered to be a risk factor for COVID-19. Drinking boiled water, as well as bathing in hot water, were considered as important preventive measures:

*"It's a slum area, people have been living here since they were born. More dangerous viruses must have come and gone here, we don't know. So, I think we are that strong. Our immunity system is strong".* (**Male, 28 years, medical representative, living in Dharavi since birth**)

"*Some people say that those who stay vegetarian, that person will not get carona. (Female, 60 years, housewife, disabled, Dharavi resident for 50 years)*

A deluge of information on COVID-19, along with facts distorted by rumours, has had repercussions in the community. On one hand, during the initial stages of the pandemic, the deluge of information from various sources seems to have contributed to intense stigma and paranoia. On the other hand, during the later stages of the pandemic, over-exposure to COVID-19 messaging along with other factors seems to have led to beliefs of non-vulnerability and lower-risk perception. These issues have been discussed further below.

**An initial phase of intense fear and panic.** When we asked people to talk about how they felt about COVID-19 during the initial stages of the pandemic (April-June 2020), everybody spoke of fear and panic. The newness of the disease and the uncertainties surrounding it had engendered intense fear of the disease. During this time period, the escalating number of cases in Dharavi was often mentioned in the news (newspaper and local television) which had added to the anxiety of people:

"*We have never heard of this carona earlier. When something new comes, one can't trust it easily, so there will be a fear of 'who knows what will happen'. (Male, 35 years, daily wage worker, years of residence in Dharavi not known)*

"*I was a bit scared, whenever I used to touch anywhere. I was very scared, I used to touch the doors with fear in my mind because I didn't know that how many people have touched it before." (Female, 27 years, housewife with a newborn, Dharavi resident for 12 years)*

Discussions with community respondents also indicated that they had been as wary of the Government's initial response to COVID-19 as of the disease itself. The government's initial strategy had involved high levels of contact tracing, screening and quarantining, and all people with Influenza Like Illnesses (ILIs) had been referred to locally set-up 'fever clinics' where they were screened for fever, oxygen levels and other underlying conditions. Further, if tested and found positive, people had been quarantined in special COVID-19 care centres set up for this purpose. This response of the government, people shared, had exacerbated fear of the disease during the initial phase of the outbreak:

"*If they go (to the hospitals) there is this fear: What if they have Covid? And what if they have a common cold only but are still kept as Covid patients in a quarantine centre. Now if anyone has a cold for 2 days, they get asked to do a covid test." (Female, 42 years, Dharavi resident for 21 years, volunteers with SNEHA)*

"*Yes, they (government) forcefully do check-ups, People say that if you come to a hospital for only cough or cold, they will show you as a corona patient." (Female, 27 years, housewife with a newborn, Dharavi resident for 12 years)*

*A woman lives a few houses away from us. She had a fever and did not go to the clinic. When her reports came positive, the police came, explained, and took her for admission. They also quarantined her family. (Male, 44 years, living in Dharavi since birth, volunteers with SNEHA*)

"*If there's fever, BMC (public hospitals) directly declares it to be Corona. Even if there is a small symptom, the entire family gets disturbed. Because of this fear, many don't go to the hospital- that if something happens to them, they (the government) will do something to the family. (Male, 50 years, social worker, Dharavi resident for 30 years)*

Due to the fear of getting "caught" with COVID-19 and being quarantined away from family, some respondents acknowledged that they had tried to hide symptoms of respiratory ailments, refrained from seeking medical help, and resorted to home remedies instead. Many people shared an underlying fear of dying away from the family in a hospital. A few respondents even shared a rumour about an injection that was being given in the hospitals that killed patients if they were found to be infected. This fear had been further fuelled by media reports of poor treatment of COVID-19 patients in hospitals, as well as the handling of dead bodies in manners that people found insensitive:

*"After that person dies, then they don't even give body to the family. So, because of this they don't go". (Female, 35 years, Dharavi resident for 15 years, volunteers with SNEHA)*

*"There was a rumour going on at that time that in hospitals they sometimes force you to take an injection that can kill you" (Male, 39 years, daily wage worker, Dharavi resident for 22 years)*

Intense fear of being infected also drove community members to discriminate against COVID-19 infected individuals. Three patients we spoke to shared experiences of discrimination; particularly of neighbours being non-helpful, unsupportive, refusing to talk to them, and 'running away' from them even after they had recovered (Table 8)

**Fear and panic have now given way to the belief that there is "No COVID-19 in Dharavi currently" (December 2020- January 2021).** When we did our interviews in December-January 2021, most people spoke of COVID-19 as a disease of the past. People reported that they did not feel as vulnerable to the disease as they did before:

*"Now everything is like earlier only. People's fear about carona is finished. Routine has started in January and February." (Male, 42 years, COVID recovered, works in a blood bank)*

*"People are still afraid, but now they say that there is no such a thing like carona. They think like that. People have started roaming around without wearing masks." (Female, 39 years, Dharavi resident for 2 years, volunteers with SNEHA)*

**Table 8. Experiences of discrimination reported by patients.**

| Case 1: Male, 39 years, sanitation worker: Contracted infection in November 2020 | Case 2: Male, 42 years, blood bank technician: Contracted infection in June 2020 | Case 3: Female, 65 years, retired community health worker: Contracted infection in August 2020 |
|---|---|---|
| He lived with his wife and two children in Dharavi, and worked as a garbage cleaner. He tested positive for COVID-19 in November 2020, and was admitted to a public quarantine facility for 10 days. He was isolated from his family and put in a government COVID-care centre. He had told his family not to share his positive status with relatives and friends. Instead, the family had told the neighbours that he had left Dharavi temporarily. When he recovered and came back, he had shared his experience with others. But he found that neighbours "ran away" from him and refused to converse with his family. | He lived with his wife and three children in Dharavi and worked as a lab technician in a blood bank. He tested positive for COVID-19 in June 2020. Since he was considered an essential worker, he was required to physically report on duty even during the lockdown. He shared that people were very scared of talking to him throughout the lockdown since he was a health worker, and maintained even more distance from his family after he got infected. | She lived with her husband, son, daughter-in-law, two grandchildren in Dharavi. Most members of her family tested positive for COVID-19 in August 2020, and were quarantined elsewhere. On their return, their neighbours refused to talk to them. She shared that many people hid their positive COVID-19 status because it was hard to deal with the discrimination faced by patients and their families after recovery. |

*"They say Corona is not there. Corona is gone now. Now there's just the common cold, which people call Corona." (Female, 35 years, Dharavi resident for 15 years, volunteers with SNEHA)*

This low perception of risk from corona has been reflected in our survey data as well (Table 5). We tried to understand why this shift in attitudes-from intense fear to almost no fear- had taken place in the Dharavi community. Most respondents denied being aware of any COVID-19 cases in their area currently. Some respondents believed that even previously, the issue had been blown out of proportion and that the surge of COVID-19 cases was never a reality in Dharavi as had been portrayed by the media:

*"Some people say the government is showing so many cases and numbers but the patients themselves have not been seen at all in Dharavi" (Male, 39 years, daily wage worker, Dharavi resident for 22 years)*

*"TV and media gave this out-of-control publicity. Through the medium of TV, they started showing such high numbers that, this many died, today's total death is this much, this much positive, this much negative. This created much fear inside people." (Male, 50 years, Social worker, Dharavi resident for 30 years)*

One of the informants shared that the fear of the disease was slowly going away from the community due to positive news of recoveries:

*Yes, now they are less scared because there's not that much news of people who pass away, Now, the media is telling good things, that people become positive but get better. That's why the fear is gone. Now they say carona is normal. Tuberculosis is harder than carona" (Male, 44 years, living in Dharavi since birth, volunteers with SNEHA)*

Further, among a few respondents, there was a belief that poor living conditions, as well as the harsh work environment in Dharavi, made the community there immune to the disease:

*First thing is, most people here are labourers. Carona cannot happen here. People living in buildings don't have to work in (poor) conditions like this, they are the ones who will definitely get this. (Male FGD participant, 35 years, utensil maker)*

Lastly, our interactions also indicated that the easing-up of lockdowns from mid-2020 onwards had signalled to the community that the pandemic was over. Thus, overall, we felt that the community in Dharavi perceived less risk from COVID-19 during the time of our interviews, and had moved on from the initial stage of intense fear and panic.

**COVID-19 preventive measures: Being practiced, but lower in intensity presently.** We had asked respondents about their practice of preventive behaviours pertaining to COVID-19 (wearing of masks, sanitization, and maintaining physical distancing), and their beliefs in the effectiveness of these practices in preventing COVID-19. Our survey findings had indicated that the majority of the respondents had reported practicing COVID-appropriate behaviours (see Table 5). Our qualitative data from phone interviews collaborated this to some extent. Due to the fear of COVID-19 during the early stages of the pandemic, it appeared that people had tried to take precautions to the best of their ability- most people spoke of wearing masks, washing hands, using sanitizers, and staying at home while limiting visits outside to buy only essentials. We also got reports of the use of some interesting home remedies to prevent COVID-19. People reported drinking hot water and herbal decoctions, increasing the use of

turmeric in their meals, and avoiding non-vegetarian food. People did not mention any problems related to water supply in their area and shared that they were able to follow hand hygiene practices. It was also generally believed that these measures were effective, though a few respondents did share that God's will or bad fate could surpass even the most diligent practice of protective behaviours against COVID-19.

However, our observations during field visits in Dharavi done in February 2021, supplemented by some phone interviews, suggested that all COVID-19 preventive practices in the community were slackening. We observed several instances of people not wearing masks. We also realised that while people reported wearing masks 'outside', they did not define the inner lanes of their neighbourhood as 'outside':

> *If you just go around your area or your neighbourhood (without a mask) then it is fine but anywhere outside we go we wear our masks at all times.* **(Male, 35 years, daily wage worker**)

> *Those who go to the road, they wear. No one wears inside the lane.* **(Female, 28 years, housewife, Dharavi resident for 7 years**)

During our in-depth discussions, people shared that of all the preventive practices pertaining to COVID-19, social distancing was the most impractical to continue presently. For one, people needed to go out to work. Further, structural inadequacies like small houses, cramped lanes, shared public toilets were mentioned as important barriers to maintaining distance within these geographical spaces:

> "*If you want to see, come here and try walking while keeping 6 feet distance. It is not possible. . .. And you know, the condition of the toilet in our community. We have to stand in the line. After one person comes out, another person goes in*" **(Male, 42 years, ward boy, currently unemployed)**

In summary, though the knowledge on preventive practices pertaining to COVID-19 was high and people generally believed in the usefulness of these practices, these were being less intensively followed at the time of this study.

In addition, we found that strong regulatory measures such as the enforcement of fines for not wearing masks or even reports of police beating people for crowding were reported as important reasons for adopting some of the COVID-appropriate behaviours during the early days of the lockdown. Once these regulatory measures were relaxed, COVID-appropriate behaviours were not sustained with the same intensity as previously:

> *The police used to beat those who were sitting together or roaming on the road. But now, this has stopped. So, people have again started sitting in a group and have started chit-chatting.* **(Female, 25 years, Dharavi resident for 8 years, volunteers with SNEHA**)

> *Everybody was at home and was following everything properly because there were many cops on the road. There was a curfew. Now everything is normal.* **(Male, 28 years, medical representative, living in Dharavi since birth**)

In summary, the general reduction in panic and fear, as well as relaxations in policing, seem to have led to less-intensive adoption of preventive practices during the time of our interviews, as compared to the earlier lockdown period.

**The story of the lockdown in 2020 and community experiences.** We found that people's experience of COVID-19 in the Dharavi community was strongly tied with the lockdown

enforced pan India (see Fig 2). The lockdown in Dharavi was reported to be stringent, with a high level of policing, travel restrictions, and intensive screening and testing for COVID-19 in the community. In fact, from the point of view of the community, the lockdown (a previously unfamiliar term and event) was associated with fear, travel restrictions, loss of jobs, and financial hardships. It was repeatedly shared by our respondents that the lockdown period was a very difficult time for people in Dharavi:

> *"I have not seen anything like this before, first time in my life I have seen a lockdown. I did not know before what was a lockdown"* **(Female, 27 years, housewife with a newborn, Dharavi resident for 12 years)**

> *"We had money problems to run the house, dad also had to leave his job, my brother also wasn't going to work. I was also at home; we had a lot of problems." ***(Male, 25 years, works in a hotel, living in Dharavi since birth)**

> *"Many people had no work. Men were sitting in their homes, there was less food, a lot of tension was there." ***(Female, 39 years, Dharavi resident for 2 years, volunteers with SNEHA)**

We discuss below some of the important repercussions of the lockdown as reported by the community.

*Migration to home-towns and back again.* The informal settlements in Dharavi have been home to many people who have migrated to the city of Mumbai seeking better earning opportunities. During the lockdown, the fear of infection, loss of jobs, inability to pay rents, and future uncertainty made many migrants leave Mumbai and return to their native places:

> *"Everyone was taking their bags and moving back to the village. Half of Mumbai city was empty. ***(Female, 60 years, housewife, disabled, Dharavi resident for 50 years)**

> *"I think most problems were faced by those giving rent. People who did not have ration card (government approved document to avail food subsidy) also, so no ration. . .and there were no relatives to help" ***(Male FGD participant, 30 years)**.

Respondents also shared numerous challenges in commuting to their native villages. (Box 2 depicts an illustrative case study of one such migrant). Since trains and buses were not running, people had to take lifts from truck drivers and other vehicles moving towards their destination point. During the time of our interviews in January 2021, it was reported that many migrants were returning to Mumbai in search of jobs again.

*Disruptions in food supply during the lockdown.* People reported that they had faced some problems in the community related to food procurement during the lockdown, but going completely without food was less reported (also see Table 6 results from the survey). It was shared that during the lockdown, many civil society organizations and community leaders stepped up to ensure that food was available for everyone. There were also reports of community kitchens, and people within the community cooking extra for those who couldn't afford meals:

> *"Sometimes we did go to buy from the shop, sometimes we got help from the temples that were distributing good grains and oil. . .sometimes different organisations would distribute food. You had to queue up and get your food. If it was over, they would get more. But no one really went hungry, which was a good thing. Somehow or the other, food was managed." ***(Male, 35 years, daily wage worker)**

### Box 2. Case study of a couple with a child (5 years) who went to their native place during lockdown

The couple lived in a rented room; the male respondent made artificial jewellery while the female respondent was a housewife. When the lockdown was announced, their sales stopped completely. For a few months family survived with the savings they had. Then, they decided to go back to the village:

*We didn't run away because of Corona fear but this was the reason. . .it was no money. We managed everything for a month or two, but after that we couldn't manage. Since we didn't have money, there was no food. . .we thought we would get some help from family at home.*

The couple borrowed some money from friends and neighbours for travelling and went in a hired truck (along with many others). They described the journey as comfortable as they got food, water, buttermilk and even glucose powder free of cost. On reaching the village, their temperature and pulse was checked and were told to quarantine for 15 days in their own house. They stayed in the village for three months.

But concerns of livelihood and pending financial commitments led the male respondent coming back to Mumbai in October 2020, while his wife and son returned in January 2021. After coming back, the couple borrowed money to start their work. Their main worry during our conversation was the economic slowdown:

*"Even now, we don't have many orders. We can just manage our expenses. . . During corona, we lost entire year. If we lose the order this year also, then we won't be able to earn anything.*

These informal support systems were spoken of highly by our respondents, despite smaller problems being mentioned in the logistics and distribution of food.

*Access to routine healthcare.* On being asked about community experiences pertaining to accessing routine healthcare during the lockdown, people shared that they did not venture out to seek treatment except for critical health ailments. People reported postponing seeking care since they were scared of contracting COVID-19 in health setups. In the case of immunization of children, most parents reported that they had postponed immunization visits to the hospitals during the lockdown until camps resumed in the community. However, people did not report major issues in accessing the nearby public hospital for institutional delivery. This could partly be because Dharavi is located very close to one of the big, public sector, tertiary care hospitals in Mumbai. In general, access to routine healthcare was not reported as a problem during the lockdown, but people's fears drove them to avoid seeking care in general.

*Gendered repercussions of the lockdown.* Our community discussions suggest that the ramifications of the lockdown might have been different for men and women. While both genders expressed financial concerns, it was shared that men needed to go out to work and the lockdown had been restrictive on men's daily routines. Women, to some extent, appeared to feel less constrained by the stay-at-home regulations, perhaps because prevailing gender norms had been restrictive about women's movements in these communities even prior to the lockdown. We encountered two instances of women telling us they were used to wearing traditional scarfs covering their faces; hence wearing masks was not a 'new' effort for them.

Women, however, did report additional household chores (cooking, cleaning) to attend to during the lockdown. There were instances of participants from both genders reporting that women were 'stronger' (resilient) than men during the pandemic, and coped better with increasing workloads and uncertainty. One of our unpublished studies from another area notes a possible increase in household stress and domestic violence against women during the lockdown [38], but we couldn't do justice to this theme in this study.

*Othering of the Dharavi community.* With the detection of the first case of COVID-19, Dharavi, being one of the most densely populated regions in Mumbai, got special attention from policymakers, public health practitioners, and subsequently the media. This spotlight on Dharavi seems to have had some unintended consequences. For one, respondents shared that the community as a whole felt 'set-aside' from the rest of the population, and people from Dharavi were treated differently just because they belonged to this geographical space. There were also reports of people from Dharavi not getting employed since employers perceived them to be of high risk to others:

> *"Listening to the name 'Dharavi', nobody gives us a job. . .even now."* **(Male FGD participant, 25 years)**

> *"My friend, she used to go to work before. I asked her why she stopped going to work, she replied that they aren't letting anyone from our area come to work. And they aren't even hiring people from here."* **(Female, 24 years, living in Dharavi since birth)**

*Overwhelming concerns about loss of livelihoods.* People reported limited employment opportunities being available currently in Dharavi. Our survey data also highlighted unemployment as the main worry. Similar concerns were reported in almost all our community discussions, and by people of both genders:

> *"Everything is closed. There is no work going on. In the area that I live in, everyone is a labourer. There is no one with a permanent job. So, if they don't work, then how will their family survive. This is the biggest question for us."* **(Female, 42 years, Dharavi resident for 22 years, volunteers with SNEHA)**

> *"The problem in the Dharavi is that the people who had small factories. . .those working as labourers, all lost their jobs. Salaries have been put on hold. The flow of money has stopped."* **(Male, 50 years, social worker, Dharavi resident for 30 years)**

In conclusion, the story of the lockdown in Dharavi suggests that concerns in Dharavi during the pandemic stretched much beyond the actual disease itself. In Fig 3, we have tried to depict the wide range of factors that emerged as important in our study in influencing health outcomes in the informal settlements of Dharavi during our study period.

Fig 3 summarizes factors that were reported by residents of Dharavi as important to their health during the pandemic. The direct response to the needs of the COVID-19 pandemic (testing and treatment) was of course of importance. But in addition to this, the need for immediate welfare services (provision of food and emergency supplies) and routine health services was strongly felt. Of long-term concern to the community were livelihoods. Our survey and qualitative interviews were conducted between October-January 2021, more than four months after the lockdown had been lifted. In this time period, people reported that the job market had not yet bounced back to the pre-COVID 19 situation. Small-scale industries were trying to recover from the losses incurred during the lockdown which led them to hire only a few people unlike earlier. There was hesitation on the part of employers to hire from Dharavi,

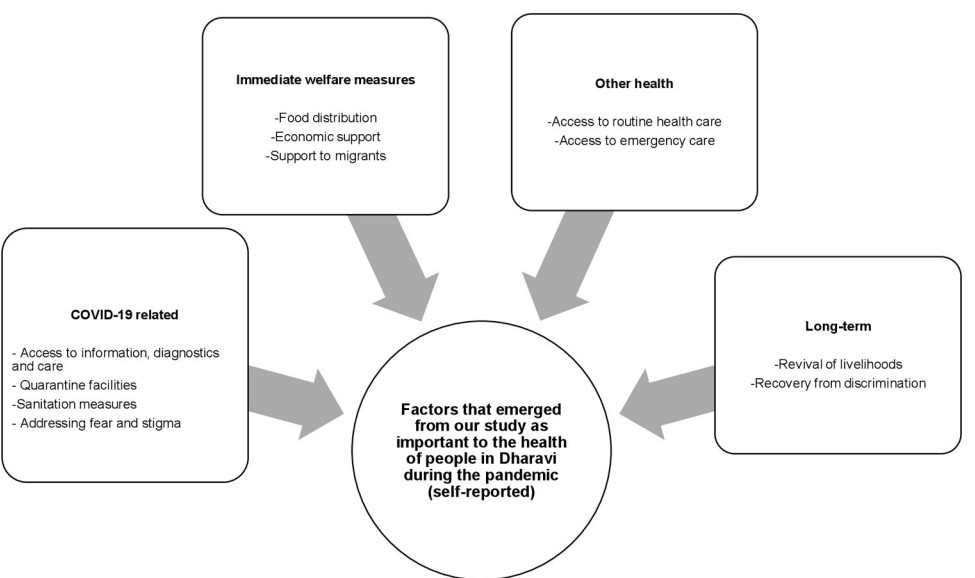

**Fig 3. Factors that emerged as influencers of health outcomes in Dharavi from our study.**

for these informal settlements had become infamous for being a COVID-19 hotspot. Overall, it was economic concerns that dominated all our conversations with the community.

## Discussion

It has been recognized that an in-depth understanding of the knowledge, attitudes, and practices of communities can help to contextualize and improve response strategies to disease outbreaks [39, 40]. Despite this understanding, literature on community perspectives on the Covid-19 pandemic, from urban informal settlements has been limited. In this study, we have attempted to provide insights into the awareness, attitudes, and reported practices of the community in Dharavi pertaining to COVID-19. We have also reported on some broader experiences and concerns of the community in Dharavi during the pandemic's first wave in 2020. One strength of this study has been the use of mixed methods, wherein quantifiable evidence has been combined with in-depth interpretative knowledge, in order to bring community experiences to the forefront.

Since this study was done during the initial stages of the pandemic, the study has some important limitations. First, knowledge about COVID-19 and its transmission had advanced from the time we administered our survey questionnaire. During the time of our survey, the airborne nature of the coronavirus transmission was not widely known. Secondly, we did the survey and the initial part of the qualitative data collection during a time when the movement of people was highly restricted. Hence, we had to collect much of our data online. We tried to take appointments and call as per the participant's convenience late in the evenings or over the weekends. But sometimes, there were issues with phone networks and connectivity. We had high rates of non-participation during data collection because it was a difficult time for people, and interest in participating in online conversations was low. In order to overcome the issue of non-participation and to minimise respondent fatigue, we had to limit the number of questions we asked in the survey and did not collect detailed demographic information including participants' education, occupation, income, and marital status. The lack of detailed demographic information limited deeper analysis in our study, but we were hoping to atleast get a

general picture of the situation in Dharavi to inform health promotion activities in the area. While cell phones are very commonly used in urban informal settlements in India (most people in Dharavi own one), we could not survey people who did not own a phone. In this way, we might have excluded certain sections of the population. Also, our findings were based on the self-reported data which does not rule out the possibility of socially desirable answers being given by respondents, leading to an overestimation of COVID appropriate behaviours. Finally, the cross-sectional survey design we used gives only a snapshot of the situation at one point in time and does not capture the evolution of knowledge and practices in the community over time.

During the qualitative interviews, while we tried our best to keep the phone interviews informal and build rapport with the participants, we could not always make meaningful connections. We also felt that the stigma against COVID-19 made people hesitant to come forward to talk to us. People often denied knowing others who had the disease, and we found very few patients (only three) who were willing to talk to us. While we were prepared for some resistance, we had not expected denial of the existence of COVID-19 in this area to be this intense. We tried to compensate for the limitations posed by phone interviews by doing some additional face-to-face interviews, FGDs, and observations when movement became possible, as noted in our methodology section.

Our findings, in general, indicate high levels of knowledge and awareness on COVID-19 in Dharavi. Such levels of knowledge have been reported from other contexts as well [11–13, 41, 42]. These high levels of awareness can be attributed to intensive messaging on COVID-19 and can be considered at least partly as a consequence of the heightened political attention given to the pandemic in national and international spheres. In our study, there were no significant differences between the responses of men and women in the survey. Unlike other studies that have reported women receiving less information on the outbreak [43], adopting preventive measures to a lower extent [44], or having higher fear scores [13], our survey did not find meaningful differences in gender-wise knowledge, attitudes, or reported practices of COVID-19. Only our qualitative findings suggested that the ramifications of the lockdown could have been different for men and women; with women reporting additional household chores to attend to even while they were seen by both genders as coping better with the changing needs of the outbreak in comparison to men.

Our findings point to a 'deluge' of information on COVID-19 and highlight the role of local television channels, family, and social media in spreading information. The findings also suggest that direct messages from the formal health system and NGOs were missing during the initial stages of the pandemic. This can be attributed at least partially to movement restrictions in this period, that prevented outreach workers from the health system and NGOs from meeting people face-to-face. Such findings have a direct implication on health promotion- for clearly, the information that has reached people through social media has not always been free of rumors and myths. Further, an overload of information through social media seems to play a role in aggravating panic and fear. Other studies have pointed this out as well [42, 45, 46]. One study has referred to COVID-19 as a 'pandemic of social media' [45]. Several concerns have been raised about the harmful effects of social media messaging across varied contexts during the pandemic [47–50].

We found reports of intense fear and panic during the early stages of the pandemic. These fears were not merely related to contracting the disease; people were scared of being quarantined and being separated from the family, of dying when away from family; of the deceased being disposed of in manners that were not culturally acceptable to the community; and of being stigmatized by neighbors if found to have the disease. These fears had many repercussions. One positive consequence was that people did report taking precautionary measures.

Our quantitative survey showed high levels of adoption of preventive practices- such as mask-wearing and hand hygiene in the community. Such high levels have been reported by other studies as well [10, 51]. We also noted several 'preventive' efforts (drinking hot water, bathing in hot water, eating only vegetarian food) made by people though these were not always in line with the medical discourse on COVID-19. A few other studies have also noted the use of home remedies and alternative medicine for COVID-19 prevention [31, 52]. Overall, however, in our study, we encountered serious attempts to wear masks, wash hands, avoid gatherings and maintain distance at least during the early stages of the pandemic.

But the fear of contracting COVID-19 also had negative consequences-one important one being that people were hesitant to come forward to get tested and access medical care- despite having adequate knowledge of symptoms of the disease. This finding has been noted by health workers in another study on COVID-19 from urban Bangalore in India [53]. In other settings, for example in the context of Ebola [54], similar hesitancies to access healthcare have been reported. In our study, the phrase that police "*catch and take away*" people was often used by respondents, and this phrase was suggestive of testing/screening for infection being done against individual wishes. Further, our interviews suggested that the fear of getting diagnosed was also rooted in the discrimination faced by patients in Dharavi. As part of our community discussions, we had spoken to three recovered COVID-19 patients, and all three reported experiencing some form of discrimination (being ignored by neighbours or told by them not to stay in the vicinity). This discrimination we saw appeared to have many similarities with the stigma faced by survivors during outbreaks such as Ebola, the stigma largely being rooted in a fear that survivors could be contagious [55, 56]. We also found that such discrimination per-sisted not only at the level of individuals but also at the level of the community at large. Indeed, at present, the entire community in Dharavi feels excluded and discriminated against. This 'othering' of certain communities during COVID-19 has been reported by another study from north India [32] as well as one study from South Africa [50]. This issue is of utmost concern since such communities were already vulnerable to mental health conditions, and the pandemic has brought with it a host of additional stressors [57–59].

Another negative consequence of high levels of fear and panic during early days and COVID-associated stigma was community-level hesitancy in the use of all routine health services. People avoided seeking routine health care out of fear of contagion and fear of being diagnosed with COVID-19 once in a hospital. The Dharavi community reported delays in accessing routine immunization services and in seeking care for most non-critical ailments. While the community itself did not perceive these delays as an issue of concern, it has been globally acknowledged that the disrupted provision of and hesitancy in the use of routine health services can contribute to poor health outcomes in the long run [6, 53, 60].

In the months of January-February 2021, during the time of our interviews, there seemed to have been a shift in attitudes towards COVID-19 and the initial fear of the disease appeared to have lessened. Some people denied the existence of COVID-19 in Dharavi (including calling the disease a conspiracy), others shared that the danger had passed and that it was time for life to get back to 'normal'. This changed attitude implied that all preventive measures against COVID were being taken less seriously in January 2021.

In January 2021, the most important concern for the community was the need to revive livelihoods. Our discussions suggested that the entire community in Dharavi currently felt excluded and discriminated against, particularly with respect to jobs. The need to establish Dharavi as 'normal' again so that jobs could be secured could be responsible to some extent for the deep denial of the existence of COVID-19 in Dharavi that we saw in our discussions. Most people at this point in time believed that the disease-related danger had passed. In comparison

to KAP studies that were done earlier in the lockdown, our study highlights less disease-related panic and anxiety, but more anxiety pertaining to the revival of livelihoods.

Indeed, the targeted nature of the intervention against Dharavi and the informal settlements being in the news constantly seems to have had an unintended adverse consequence on livelihoods in the community. Urban informal settlements are often dependent mainly on micro-enterprises and informal labour; and the extended lockdown in Dharavi combined with the persisting stigma against the community living in this area seems to have only increased the economic vulnerability of its residents. Such unintended consequences have been seen in other settings as well; one analysis in particular notes that measures to control outbreaks can cause as much shock as the outbreak itself and 'backfire' if adequate steps for the social protection of populations are not taken at the appropriate time [61].

The findings of this study suggest that urban informal settlements have a range of deeply contextualized requirements to combat outbreaks like COVID-19. We discuss three of these also refer to Box 3).

One, there is a need for clear messaging in the community about COVID-19 in the community. We felt that the community in Dharavi needs access to factual as well as practical information (where to go, what numbers to contact, what to do in an emergency) through means other than television or social media, conveyed in a manner that does not engender stigma, panic or denial of the existence of COVID-19. Currently, there is some dissonance between the communication that had been done so far and what the community requires. Messages that can balance both extremes of paranoia and complete denial, and counter stigma and misinformation seem to be the need of the hour.

Secondly, since COVID-19 is being thought of as a past danger, preventive measures in the community are being taken less seriously in February 2021. There is a need to put in place checks and balances (like the local regulation of mask-wearing), embed good practices pertaining to community sanitation (like the regular disinfection of public toilets), and continue to advocate social distancing in manners that are acceptable to the community (in local shops, clinics, religious places and other places where people gather in the community). As a preventive measure, the notion of social distancing needs a special mention. Despite people reporting very high rates of social distancing in our survey, probing on this topic during our interviews yielded several reports of the inability of the community to do so in meaningful ways due to the dense population, lack of space, and shared sanitation facilities in the slums. Such challenges due to social distancing have been pointed out by other studies as well; and it has been highlighted that such notions are "bewildering" to residents in informal settlements and that it was impossible in such crowded settings to even maintain the "veneer of social distance" [62, 63]. These findings re-emphasize the need to contextualize solutions such as social distancing to the unique realities of the urban informal settlements.

Lastly, while the strongly enforced response by the government and the 'flattening of the COVID-19 curve' in this area must be commended, this study clearly brings out that such a manner of response has had unintended adverse consequences in the community. The lockdown has had a deep and lasting impact on the lives of people in Dharavi and the informal economy that sustains the place (leather, food, garments, imitation jewellery) seems to have got affected. There is currently an overwhelming concern about livelihoods in Dharavi, which calls for a range of long-term social protection measures beyond immediate economic relief.

There are some wider learnings on pandemic response in general from this study. It is important for governments to keep in mind that mitigating the immediate risks of a pandemic is only the first step of a 'successful' response. Our study re-emphasizes that the recovery of communities from disasters is a slow process and needs long-term support. In the case of Dharavi, we seem to have achieved short-term success due to intensive efforts pertaining to

Box 3. Beyond clinical intervention: What does Dharavi need in the aftermath of the COVID-19 wave (2020)

## Messaging

Clear messaging on the disease through trusted sources

Messages that have practical information

Messages that work to counter stigma, misinformation, and denial

## Reinforcement of COVID-appropriate behaviors

Local regulation of behaviors through community leaders and volunteers

Continued good practices like sanitization of public spaces

Continued low-key surveillance on COVID-19

Regulation of physical distancing in places where people gather (shops, clinics, religious places)

## Economic relief

Immediate relief work to be taken up through government and non-government partnerships

Long-term social protection measures

testing, extremely complex quarantine logistics, screening, and policing must be commended. Positive acknowledgments of the efforts of the government and civil society were of course heartening to see. However, our findings suggest that this is only the first step in the response. There is a need for a culturally congruous ongoing response, that is wider in scope, even under conditions of low disease transmission. Government responses to epidemics need to be holistic, far-reaching, and stretch beyond immediate clinical intervention and disaster relief.

## Supporting information

**S1 File. Qualitative tool.**
(DOCX)

**S2 File. Quantitative tool.**
(DOCX)

## Acknowledgments

We are grateful to all donors of the 'Mission Dharavi' program for their support in implementation. We acknowledge all the participants of this study for sharing their views and experiences with us. We also appreciate the hard work done by SNEHA staff and community volunteers during this pandemic. Finally, we are thankful to Vanessa D'souza, Archana Bagra and members of the SNEHA Research Group for their valuable inputs into this study.

## Author Contributions

**Conceptualization:** Sudha Ramani, Manjula Bahuguna, Sushma Shende, Anagha Waingankar, Nikhat Shaikh, Sushmita Das, Shanti Pantvaidya, Armida Fernandez, Anuja Jayaraman.

**Data curation:** Apurva Tiwari, Rama Sridhar, Nikhat Shaikh, Sushmita Das.

**Formal analysis:** Sudha Ramani, Manjula Bahuguna, Apurva Tiwari, Rama Sridhar, Nikhat Shaikh.

**Funding acquisition:** Anuja Jayaraman.

**Methodology:** Sudha Ramani, Manjula Bahuguna, Apurva Tiwari, Anuja Jayaraman.

**Project administration:** Manjula Bahuguna, Sushma Shende, Anagha Waingankar, Anuja Jayaraman.

**Supervision:** Sudha Ramani, Sushma Shende, Anagha Waingankar, Sushmita Das, Shanti Pantvaidya, Armida Fernandez.

**Validation:** Sudha Ramani, Apurva Tiwari, Rama Sridhar, Armida Fernandez.

**Writing – original draft:** Sudha Ramani, Manjula Bahuguna, Anuja Jayaraman.

**Writing – review & editing:** Sudha Ramani, Manjula Bahuguna, Sushma Shende, Anagha Waingankar, Rama Sridhar, Nikhat Shaikh, Sushmita Das, Shanti Pantvaidya, Armida Fernandez, Anuja Jayaraman.

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
