## [Decision Letter · Decision Letter 0]

9 Feb 2022

PONE-D-21-19786Corona was scary, lockdown was worse : A mixed-methods study of community perceptions on COVID-19 from urban informal settlements of MumbaiPLOS ONE

Dear Dr. Jayaraman,

Thank you for submitting your manuscript to PLOS ONE. After careful consideration, we feel that it has merit but does not fully meet PLOS ONE’s publication criteria as it currently stands. Therefore, we invite you to submit a revised version of the manuscript that addresses the points raised during the review process. The reviewers found this study to have strong scientific merit, and made some suggestions of how to strengthen the study. Please see the detailed comments below. In particular, there is a need for a stronger description of the sampling technique used. For example, how was the phone list obtained and how representative is this phone list of the population of Dhavari? Please consider adding in a more detailed description of the demographics of the population of Dhavari as well as a description of the COVID-19 situation in the community. Please consider addressing the possible limitation of not asking more demographic questions of participants. Reviewers also suggest a deeper discussion linking the qualitative findings of the study to health risks/outcomes, and, if possible to consider revising Figure one to add more relevant information such as impacts of lockdown related to qualitative findings.

We look forward to receiving your revised manuscript.

Kind regards,

Samantha C Winter, Ph.D.

Academic Editor

PLOS ONE

Journal Requirements:

Furthermore, when reporting the results of qualitative research, we suggest consulting the COREQ guidelines: http://intqhc.oxfordjournals.org/content/19/6/349. In this case, please consider including more information on the number of interviewers, their training and characteristics; and please provide the interview guide used.

3. Thank you for stating in your Funding Statement: "This research was partially funded through the Epic Foundation. The funders had no role in study design, data collection and analysis, decision to publish, or preparation of the manuscript."

Reviewers' comments:

Reviewer's Responses to Questions

**Comments to the Author**

1. Is the manuscript technically sound, and do the data support the conclusions?

Reviewer #1: Yes

Reviewer #2: Yes

2. Has the statistical analysis been performed appropriately and rigorously? 

Reviewer #1: Yes

Reviewer #2: N/A

3. Have the authors made all data underlying the findings in their manuscript fully available?

Reviewer #1: Yes

Reviewer #2: No

4. Is the manuscript presented in an intelligible fashion and written in standard English?

Reviewer #1: Yes

Reviewer #2: Yes

5. Review Comments to the Author

Reviewer #1: This is an interesting and important mixed methods study. The original work is diligent with recruitment of participants and the methods section is very clear. The results of the survey are less interesting in terms of perceptions and knowledge of COVID-19 risks and protective practices. The reality is the global scientific community is still uncertain about much of this, so we wouldn't expect community residents anywhere to express a consensus. An important, perhaps under-emphasized, dynamic of the survey was that a majority of respondents were <30 years, the high trust in 'electronic media,' and low trust in health systems. I would encourage the authors to expand on this finding as it relates to community-level health promotion.

The qualitative data are much stronger and insightful than the survey. The quotes are powerful and communicate the lived impact of COVID-19 policies and practices, as well as the existing vulnerabilities residents of Dharavi faced before COVID. The paper's attention to discrimination, food insecurity and gendered impacts from lockdowns/COVID policy are revealing. However, the paper needed to link these and related qualitative findings to health risks and outcomes. The discussion section relates some of the qualitative findings to the literature on health inequities/social determinants of health for slum dwellers, but this would make the paper more applicable this important and under-studied area of public health. For instance, the discrimination - stress - immune system/disease links are well known, but not frequently discussed for slum populations. This would also ensure the policy/practice implications of the study address fundamental health determinants, from jobs to food to housing, not just emergency responses.

Figure 1 doesn't add any value to the research study. A map of the location of the study and images of lockdown impacts, especially as they relate the qualitative findings would add to the findings.

Reviewer #2: This is a highly relevant and well conducted study that should be published. I have the following major comments.

I lack background information on demographics for Dharavi. Like what is the age profile? This must be important as more old people gets infected. It would also help in assessing if the 468 responders represent the entire population in a meaningful way.

I also lack more information on the general COVID-19 situation in Dharavi. The text is mentioning 2543 cumulative cases in July 2020 (ref 18) but there must have been far more cases than that. How many people died? How many people were moved to quarantine outside the area? What happened in the subsequent waves? When were vaccines introduced, they are not mentioned in the text?

I am concerned about the use of cell phones for interview as I would think that there must be lots of people who can’t afford to own a phone. It is also unclear how the initial telephone list was obtained and to which degree the list is representative for the sampling universe of the 15.000 households and Dharavi in general. It is described how demographic information including participant’s education, occupation, income and marital status was not sampled due to concern about respondent fatigue, but this is also a weakness in the study.

Minor comments

242 Similarly, the tool had options like

253 What about ventilation? COVID-19 is an airborne virus.

299 Isn’t is very low that only 2 in 5 agreed. Should this be elaborated more in “discussion”.

312 Table 1: The table could be more systematic. The demographic details of participants should be the same in all cases. The question “Underlying fear and mistrust of Government response” seems to be biased, why not “Trust in Government response”

374. Table 2. How does the sample compare to the general demographics of Dharavi?

383. Table 3. It seems like the trust in NGO’s, public events and community leaders is quite limited. This issue should go more directly into “discussions”

412. It seems very optimistic that 77.1% of reusable cloth mask are washed after every use. How is this defined? Washing every day, every week or every single time the mask has been on/off?

427. Table 6. Use the word COVID-19 instead of COVID and Coronavirus

631. It would be better if people would wear masks in the narrow inner lanes.

739. Nobody seems to be reporting fear of losing property if quarantined outside their home?

6. PLOS authors have the option to publish the peer review history of their article (what does this mean?). If published, this will include your full peer review and any attached files.

Reviewer #1: No

Reviewer #2: No

---

## [Author Response · Author response to Decision Letter 0]

26 Mar 2022

RESPONSE

Dear Editors

We thank you for your valuable time and input into our paper. The peer-review was very useful. We thank the reviewers for acknowledging the merit of the paper as well as the validity of the methods we used. 

Based on the comments that we have received, we have made the following changes to the manuscript

• Added a stronger description of the sampling details in the quantitative methods section of the paper. 

• We have also added some background demographic details and a description of the COVID-19 situation in the Dharavi community (see figure 2 and box 1).

• To the limitations section, we have added some points on the limitations of our sampling methods and high non-participation rates. Particularly, we have tried to explain why we had to resort to surveying people online. 

• It was a good suggestion from reviewer 1 to have a summarize qualitative findings with respect to the health outcomes/risks, and we have weaved in some of this information into the manuscript findings and discussion sections. We have added another figure (figure 3) to depict the summarized findings.

• We have also revised table 1 and table 5 as per comments from the reviewers.

• We have appended the tools for the study as requested by the reviewers

More detailed response to the reviewers’ points has been given below.

We have also included a revised funding statement: This research was funded through the Epic Foundation and Give Foundation, as a part of larger implementation grants. The funders had no role in study design, data collection and analysis, decision to publish, or preparation of the manuscript.

As per the data sharing policies of our organization, SNEHA, program or project datasets can be shared online only after three years of completion of the program or project. However, we are happy to share data individually if people write to us. The applicant must have clearance from their local ethics board before accessing our data. All SNEHA datasets are anonymised or pseudonymised. No identifiers of participants will be shared under any circumstances. 

Looking forward to hearing back from you.

Warm regards

Dr. Anuja Jayaraman

Journal Requirements:

We have included a copy of the questionnaire.

Furthermore, when reporting the results of qualitative research, we suggest consulting the COREQ guidelines: http://intqhc.oxfordjournals.org/content/19/6/349. In this case, please consider including more information on the number of interviewers, their training and characteristics; and please provide the interview guide used.

We have reported qualitative information according to standard guidelines and also included information on the interviewers. We have also attached the interview guide that we used.

3. Thank you for stating in your Funding Statement: "This research was partially funded through the Epic Foundation. The funders had no role in study design, data collection and analysis, decision to publish, or preparation of the manuscript."

We have now revised the funding statement to read

This research was funded through the Epic Foundation and Give Foundation, as a part of larger implementation grants. The funders had no role in study design, data collection and analysis, decision to publish, or preparation of the manuscript.

As per the data sharing policies of our organization, SNEHA, program or project datasets can be shared online only after three years of completion of the program or project. However, we are happy to share data individually if people write to the corresponding author. The applicant must have clearance from their local ethics board before accessing our data. All SNEHA datasets are anonymised or pseudonymised. No identifiers of participants will be shared under any circumstances. 

We are happy to provide you with an official copy of our data sharing policies if needed. Thank you for understanding.

We have rechecked all our references. Thank you again.

Comments to the Author

1. Is the manuscript technically sound, and do the data support the conclusions?

Reviewer #1: Yes

Reviewer #2: Yes

2. Has the statistical analysis been performed appropriately and rigorously?

Reviewer #1: Yes

Reviewer #2: N/A

3. Have the authors made all data underlying the findings in their manuscript fully available?

Reviewer #1: Yes

Reviewer #2: No

4. Is the manuscript presented in an intelligible fashion and written in standard English?

Reviewer #1: Yes

Reviewer #2: Yes

5. Review Comments to the Author

RESPONSE TO REVIEWERS

Reviewer #1: This is an interesting and important mixed methods study. The original work is diligent with recruitment of participants and the methods section is very clear. The results of the survey are less interesting in terms of perceptions and knowledge of COVID-19 risks and protective practices. The reality is the global scientific community is still uncertain about much of this, so we wouldn't expect community residents anywhere to express a consensus. An important, perhaps under-emphasized, dynamic of the survey was that a majority of respondents were <30 years, the high trust in 'electronic media,' and low trust in health systems. I would encourage the authors to expand on this finding as it relates to community-level health promotion.

The qualitative data are much stronger and insightful than the survey. The quotes are powerful and communicate the lived impact of COVID-19 policies and practices, as well as the existing vulnerabilities residents of Dharavi faced before COVID. The paper's attention to discrimination, food insecurity and gendered impacts from lockdowns/COVID policy are revealing. However, the paper needed to link these and related qualitative findings to health risks and outcomes. The discussion section relates some of the qualitative findings to the literature on health inequities/social determinants of health for slum dwellers, but this would make the paper more applicable this important and under-studied area of public health. For instance, the discrimination - stress - immune system/disease links are well known, but not frequently discussed for slum populations. This would also ensure the policy/practice implications of the study address fundamental health determinants, from jobs to food to housing, not just emergency responses. Figure 1 doesn't add any value to the research study. A map of the location of the study and images of lockdown impacts, especially as they relate the qualitative findings would add to the findings.

Dear Reviewer 1,

We thank you for your interest in the study. Your acknowledgment of its importance and clarity of methods used is appreciated. We have noted all your suggestions and have tried our best to address these as bulleted below

Uncertainty and lack of consensus: You are right in pointing out the lack of consensus in perceptions of COVID-19. The quantitative survey was done during a time of much uncertainty, during the initial stages of the pandemic. While we adapted from an existing questionnaire, there has been much progress since then in the understanding of the virus, its transmission, and protective practices. Since this is an important point, we have now added a sentence on this to the limitation sub-section in the discussion.

Dharavi context: We have tried to provide better context to the survey findings by adding details of the COVID-19 situation in Dharavi to the introduction. This is intended to better situate the survey findings and hopefully make these findings more interesting to the reader.

Survey findings: The two survey findings pointed out by you have been noted and elaborated on in the revised manuscript. We have now revised table 3 on sources of information on COVID-19 to better represent the trust percentages and highlighted some of the key points. We realised that the previous table might get misinterpreted. The main source of information reported was electronic media, 90.2% reported television and radio followed by family or friends (56.4%) and social media (47.4%). Health system workers and NGO workers were mentioned by fewer respondents as sources of information on COVID-19. However, we also asked a question on which sources of information were most trusted by people. In response to this question, we found that the trust reported by respondents on information received from health systems and NGO workers was high (96-97%). At the same time, nearly 60% of the people reported trusting the information they obtained from social media.

Again, the time of our survey might have played a role in the nature of these findings- for NGO workers as well as outreach workers from the formal health system were constrained in their ability to provide information due to movement restrictions. Clearly, information from social media-even when acknowledged as unreliable- has played a key role in shaping people’s perceptions of the pandemic. As you pointed out, the above points have important implications for health promotion- and we have added a note on these in the revised discussion section as well.

Qualitative findings: Thank you for appreciating our qualitative findings. We have tried to compile quotations that gave readers a flavour of the field- situation. In the revised version of the paper, we have tried to link some of the qualitative findings to health outcomes. While direct, ‘causal’ links are not possible to extract from qualitative results, we have tried to elaborate on some of the factors that people reported as important to their health during the pandemic. We have particularly emphasized the fact that some of these factors were directly related to COVID-19, but others have to do with access to healthcare for other conditions, need for welfare, and revival of livelihoods. We have added a new figure to depict these factors (see figure 3 in the revised manuscript)

We have also added a few papers on stress in urban informal settlements and the pandemic to the discussion section- since, as pointed out, mental health during the pandemic has been an important cause of concern. Thank you for suggesting this line of thinking.

Figure 1 and location of the study: We have now revised figure 1 to depict COVID-19 cases in Dharavi and the dates of the lockdown in India. We have also added a map of the location of the study to the paper.

Reviewer #2: This is a highly relevant and well conducted study that should be published. I have the following major comments. I lack background information on demographics for Dharavi. Like what is the age profile? This must be important as more old people gets infected. It would also help in assessing if the 468 responders represent the entire population in a meaningful way.

Dear Reviewer

Thank you for your comments and we acknowledge and appreciate your appraisal of our manuscript. We have noted your suggestions and tried our best to address these. Thank you for pointing out the lack of background information on Dharavi. We realise that this information is important for a reader who is not familiar with the Dharavi context. We have added some details on the demographics of Dharavi to our paper. We have also added some generic details on Dharavi to enable the reader visualise the informal settlements better. 

I also lack more information on the general COVID-19 situation in Dharavi. The text is mentioning 2543 cumulative cases in July 2020 (ref 18) but there must have been far more cases than that. How many people died? How many people were moved to quarantine outside the area? What happened in the subsequent waves? When were vaccines introduced, they are not mentioned in the text?

Thank you for pointing out these issues. These are indeed important details that help to set the context to the study. We have now added the following to the paper-

-Figure 1 shows the location of Dharavi. 

- Figure 2 notes cases and death count of COVID-19 in Dharavi, and depicts when the peaks occurred. We realise that this data is needed to set the context to our paper.

-We have also added some details on the response to COVID-19 in Dharavi in Box 1 including quarantine arrangements and other details that might be relevant. 

- During our data collection phase, vaccination had not started in Dharavi. Vaccinations began for the general population in March 2021 in a phased manner. We have added this sentence into the methods section.

I am concerned about the use of cell phones for interview as I would think that there must be lots of people who can’t afford to own a phone. It is also unclear how the initial telephone list was obtained and to which degree the list is representative for the sampling universe of the 15.000 households and Dharavi in general. It is described how demographic information including participant’s education, occupation, income and marital status was not sampled due to concern about respondent fatigue, but this is also a weakness in the study. 

Thank you for this valuable comment. While cell phones are very commonly used in urban informal settlements (most people own one), this point has been of concern to us as well. We were not able to access people who did not own a phone, and this is an important limitation of our study. Further, we faced other issues too- for instance, bad connectivity and networks and challenges in getting hold of people for the interviews. We have now elaborated on some of these issues in our limitation section of the study explicitly.

In the limitation section, we have also tried to explain why we had to resort to this way of sampling. We did the quantitative study during the peak lockdown in India (Sept 2020), when the movement of people was highly restricted. The risk of infection was also very high and physical data collection was impossible. However, our qualitative study was done a little later (November 2020-Jan 2021). We did the initial round of qualitative data collection over the phone. When the lockdown had lifted and it was safe to travel, we supplemented the phone interviews with face-to-face interviews particularly sampling people who were not accessible by phone. In this manner, we tried to make up for some of the limitations of the initial data collection through the phone. We do agree with you that this was a limitation of our study and we have tried to explicitly explain this. Our inability to collect detailed demographic information was indeed a limitation and we have noted this in the revised paper.

We have also added more information in the text to clarify the process through which we obtained phone numbers. We hope that these revisions help the reader interpret our findings in line with the limitations.

Minor comments

242 Similarly, the tool had options like

Added

253 What about ventilation? COVID-19 is an airborne virus.

During the time of our survey, knowledge of COVID-19 and its spread was still evolving. The questionnaire we had adapted from did not have information on ventilation. We have added a small note on this to table 5. However, we did ask one question to assess respondent’s knowledge about the spread of COVID-19 through air (see table 4).

299 Isn’t is very low that only 2 in 5 agreed. Should this be elaborated more in “discussion”.

Our survey had high rates of non-participation, particularly because it was a difficult time for people and interest in participating in online surveys was low. We have added this point and discussed this more in the limitation section of our study.

312 Table 1: The table could be more systematic. The demographic details of participants should be the same in all cases. The question “Underlying fear and mistrust of Government response” seems to be biased, why not “Trust in Government response”

Thank you for pointing this out. We have now revised this table to read better.

374. Table 2. How does the sample compare to the general demographics of Dharavi?

As per the Census 2011 data, the majority (70%) of the Indian population is young and belongs to the working-age group. Data also indicates that urban areas have less elderly (29%) population in comparison to the rural areas (71%). According to the High Powered Expert Committee 2011, direct migration to urban areas accounts for 20 to 25% of the increase in urban population. Dharavi which is a hub for small-scale industries attracts young migrants from all over the country for livelihood opportunities. Most of the participants in our study were in working-age group of 18 to 40 years which is representative of Dharavi’s young and working population in general. We have now added these notes to the paper, to give the reader a better sense of our sample. Thank you for this comment.

383. Table 3. It seems like the trust in NGO’s, public events and community leaders is quite limited. This issue should go more directly into “discussions”

Thank you for pointing this out. Our other reviewer pointed this out as well. We have now revised table 3 on sources of information on COVID-19 to better represent the trust percentages and highlighted some of the key points. We realised that the previous table could be misinterpreted. The main source of information reported was electronic media, 90.2% reported television and radio followed by family or friends (56.4%) and social media (47.4%). Health system workers and NGO workers were mentioned by fewer respondents as sources of information on COVID-19. However, we also asked a question on which sources of information were most trusted by people. In response to this question, we found that the trust reported by respondents on information received from health systems and NGO workers was high (96-97%). At the same time, only 60% of the people reported trusting the information they obtained from social media.

Again, the time of our survey might have played a role in the nature of these findings- for NGO workers as well as outreach workers from the formal health system were constrained in their ability to provide information due to movement restrictions. Clearly, information from social media-even when acknowledged has unreliable- has played in key role in shaping people’s perceptions of the pandemic.

412. It seems very optimistic that 77.1% of reusable cloth mask are washed after every use. How is this defined? Washing every day, every week or every single time the mask has been on/off?

Thank you for pointing this out. 77.1% of the people responded that they washed their reusable cloth masks after every use. As this is self-reported data, we cannot rule out the possibility of socially desirable answers. To avoid confusion, we have now removed this data point from the text.

427. Table 6. Use the word COVID-19 instead of COVID and Coronavirus

Revised.

631. It would be better if people would wear masks in the narrow inner lanes.

Thank you for sharing your concern. It was interesting to note that people did not think of the narrow lanes as ‘outside’ of their homes.

739. Nobody seems to be reporting fear of losing property if quarantined outside their home?

This is true. Entire families were usually not quarantined. Only the affected individuals were quarantined. We have added some details in Box 1 to clarify this.

Thank you for this review and regards.

---

## [Editor Report · Decision Letter 1]

25 Apr 2022

Corona was scary, lockdown was worse : A mixed-methods study of community perceptions on COVID-19 from urban informal settlements of Mumbai

PONE-D-21-19786R1

Dear Dr. Jayaraman,

We’re pleased to inform you that your manuscript has been judged scientifically suitable for publication and will be formally accepted for publication once it meets all outstanding technical requirements.

Kind regards,

Samantha C Winter, Ph.D.

Academic Editor

PLOS ONE
---

## [Editor Report · Acceptance letter]

28 Apr 2022

PONE-D-21-19786R1 

Corona was scary, lockdown was worse: A mixed-methods study of community perceptions on COVID-19 from urban informal settlements of Mumbai 

Dear Dr. Jayaraman:

I'm pleased to inform you that your manuscript has been deemed suitable for publication in PLOS ONE. Congratulations! Your manuscript is now with our production department. 

Kind regards, 

on behalf of

Dr. Samantha C Winter 

Academic Editor

PLOS ONE